# Role of outer surface probes for regulating ion gating of nanochannels

Xinchun Li[1,2], Tianyou Zhai [1], Pengcheng Gao[3], Hongli Cheng[1], Ruizuo Hou[1], Xiaoding Lou[1,3] & Fan Xia[1,3]

Nanochannels with functional elements have shown promise for DNA sequencing, single-molecule sensing, and ion gating. Ionic current measurement is currently a benchmark, but is focused solely on the contribution from nanochannels' inner-wall functional elements (NIWFE); the attributes of functional elements at nanochannels' outer surface (NOSFE) are nearly ignored, and remain elusive. Here we show that the role of NOSFE and NIWFE for ion gating can be distinguished by constructing DNA architectures using dual-current readout. The established molecular switches have continuously tunable and reversible ion-gating ability. We find that NOSFE exhibits negligible ion-gating behavior, but it can produce a synergistic effect in alliance with NIWFE. Moreover, the high-efficiency gating systems display more noticeable synergistic effect than the low-efficiency ones. We also reveal that the probe amount of NOSFE and NIWFE is almost equally distributed in our biomimetic nanochannels, which is potentially a premise for the synergistic ion-gating phenomena.

[1] State Key Laboratory of Material Processing and Die & Mould Technology, School of Material Sciences and Engineering, Hubei Key Laboratory of Bioinorganic Chemistry & Materia Medica, School of Chemistry and Chemical Engineering, Huazhong University of Science and Technology (HUST), 430074 Wuhan, China. [2] Pharmaceutical Analysis Division, School of Pharmacy, Guangxi Medical University, 530021 Nanning, China. [3] Faculty of Materials Science and Chemistry, China University of Geosciences, 430074 Wuhan, China. Xinchun Li and Tianyou Zhai contributed equally to this work. Correspondence and requests for materials should be addressed to F.X. (email: xiafan@hust.edu.cn)

Transmembrane ion channels in living organisms can regulate the transport of ions and molecules under external stimuli, which is vital to life process[1–3]. This process is generally executed in a cooperative fashion by two-channel subunits that are composed of protein functional elements delicately organized at the outer and inner of ion channels[4,5], which fulfill gating function through conformational change. To mimic the response-to-stimulus behaviors of biochannels, diverse artificial nanochannels have been fabricated; for example, ion[6], pH[7], light[8], and temperature[9]-activated nanochannels and dual-stimuli responsive nanochannels including pH/temperature[10], pH/light[11], pH/molecule[12], and pH/voltage[13] have been intensively studied. It is worth mentioning that the current work is nearly focused on the contribution from the functional elements at nanochannels' inner wall (NIWFE)[6,7,10,11,13–19]; the contribution of the functional elements on nanochannel outer surfaces (NOSFE) has been practically ignored. Molecular theory and simulation calculation have been proposed to predict ionic conductance and gating behavior[20,21]; however, experimental investigations are yet unfulfilled.

A critical question thus arises: what role do regional functional elements play in ion gating? Inability to clarify this issue has long been a barrier to understanding the delicate function of biomimetic nanochannels. In addition, this is also a problem in nanochannel-based DNA sequencing[22–29] and single-molecule sensing[30–36]. For instance, discriminating a nucleotide base typically relies on characteristic channel-blockade current and dwell-time signatures, which, however, cannot be actualized when the base has no access to the inner of nanochannel. It thus may be envisioned that rationally structured NOSFE would be possibly beneficial to differentiate a specific base in a DNA sequence. Likewise, recognition of a single molecule is generally accomplished when the target molecule is captured by functional elements oriented at the inner of nanochannel, through which target molecule recognition can be evolved to a sensing event and translated to characteristic ionic current (IC) signal. Detection of individual DNA abasic sites[37], investigation of single-molecule stereochemistry reaction[38], and probing single-molecule enzyme kinetics[31] have been realized in this manner. This sensing strategy also may be improved provided that NOSFE is taken into account, since it can prescreen a target molecule at the nanochannel outer surface rather than rely solely on the NIWFE, which shall enhance sensing-specificity and flexibility. All these concerns, however, remain to date untapped. The present work attempts to reveal how NOSFE regulates ion gating. To achieve this, we construct biomimetic solid-state nanochannels using DNA-architectures-based functional elements that can sensitively and specifically respond to external stimulus, for example, ATP molecule. The gating efficiency is experimentally elucidated with integrated dual-current signals, which offer a technical means to probing of ion-gating behaviors of biomimetic nanochannels.

## Results

### Experimental design and signal recording.
We begin with metal modification of porous anodic aluminum oxide (AAO, 40–70 nm pore size) membranes[39] by controllably depositing gold (or plus titanium) layers using electron-beam evaporation technique, which produces regionally metal-decorated nanochannels (Fig. 1a, Supplementary Fig. 1), thereby forming distinct regions for oriented assembly (including inner-wall assembly, outer-surface assembly, inner-wall and outer-surface assembly) of DNA functional elements via thiol chemistry and base-pairing rule (Fig. 1b). Gating efficiency is a crucial merit of biomimetic nanochannels and is generally defined as $I_{on}/I_{off} \times 100\%$ ($I_{on}$ is IC

measured at the opening state of channel, and $I_{off}$ is IC measured at the closed state of channel).

To systematically study regional ion-gating function of biomimetic nanochannels, we investigate a low-efficiency ion-gating system composed of DNA traditional sandwich (DNA TS) and a high-efficiency ion-gating system created from DNA superstructure (DNA SS), both labeled with double-methylene blue (MB) electroactive tags (Supplementary Fig. 2). The assembled DNA frameworks act as molecular switch mediating electron transfer (ET) to generate electrolytic current (EC) signal, as well as guide transmembrane ions transport (IT) to produce IC signal, both of which are monitored by a home-made dual-current electrochemical setup (Fig. 1c).

### Probing of low-efficiency ion-gating systems.
A DNA TS structure containing ATP-aptamer sequence and MB electrochemical tags, and the disassembly process, was schematically described (Fig. 2a). The immobilization of capture probe (CP), a part of NIWFE in the nanochannels, resulted in decreased IC response, confirming the successful modification of CP in the nanochannels (Supplementary Fig. 3). We first employed cyclic voltammetry (CV) to examine the ET feature of the MB-labeled DNA TS system. The obviously reduced EC response after binding to ATP verified the successful construction of the molecular switch (Supplementary Fig. 4), since ATP-triggered disassembly of DNA TS structure led the signal probes (SP1 and SP2) away from the gold surface; consequently, the ET process was weakened. We then used square-wave voltammetry (SWV) to investigate the molecular switch features. As detailed in Fig. 2b–d, the reduction current change due to ATP-triggered disassembly varied at different regions of the nanochannels. A 62.2% (±4.1%) signal change by DNA TS-based NIWFE (Fig. 2b) can be seen, while 74.8% (±6.1%) signal change was obtained at the NOSFE (Fig. 2c), manifesting favorable molecular switch property. The relatively higher signal-off ratio of NOSFE was possibly ascribed to more accessible recognition of ATP molecules and consequently larger EC change in the bulk solution phase, which was further confirmed by examining DNA-based NIWFE + NOSFE, since the signal-off ratio was found to be 68.8% (±4.4%) (Fig. 2d). In all of the three cases, our DNA TS-based nanochannels could work well with respect to the molecular switches through EC signal characterization (Fig. 2e), which thus laid a premise to investigating the ion-gating function.

The NIWFE blocked the avenue of ions transport and generated restrained IC signal (Fig. 2f), whereas the recognition of ATP brought about amplified IC signal (with 2.2-fold current increase, measured at ±0.2 V), due to ATP-induced dismantlement of the DNA sandwich structure, and thus the enhancement of transmembrane ions flux. By contrast, the biomimetic nanochannels showed nearly negligible ion-gating effect, provided that the same DNA functional elements was just located at the outer surface of the nanochannels (NOSFE); the binding of ATP merely caused 12.6% (±3.2%) current change (Fig. 2g). However, the NIWFE + NOSFE resulted in about 2.5-fold signal-on ratio (Fig. 2h), indicating that the outer-surface probes can synchronously increase the ion-gating capacity. This enhancement phenomenon can be ascribed to a mingled effect of the outer-surface and inner-wall functional elements that cooperatively formed the ion gate[40]. Note that in the present DNA TS systems, the ion gates took effect with a low-efficiency mode in view of the IC signal-on ratios, regardless of the regional distribution of the assembled DNA functional elements at the nanochannels (Fig. 2i). This was probably due to the limited structure dimensions of DNA TS architectures confined at the

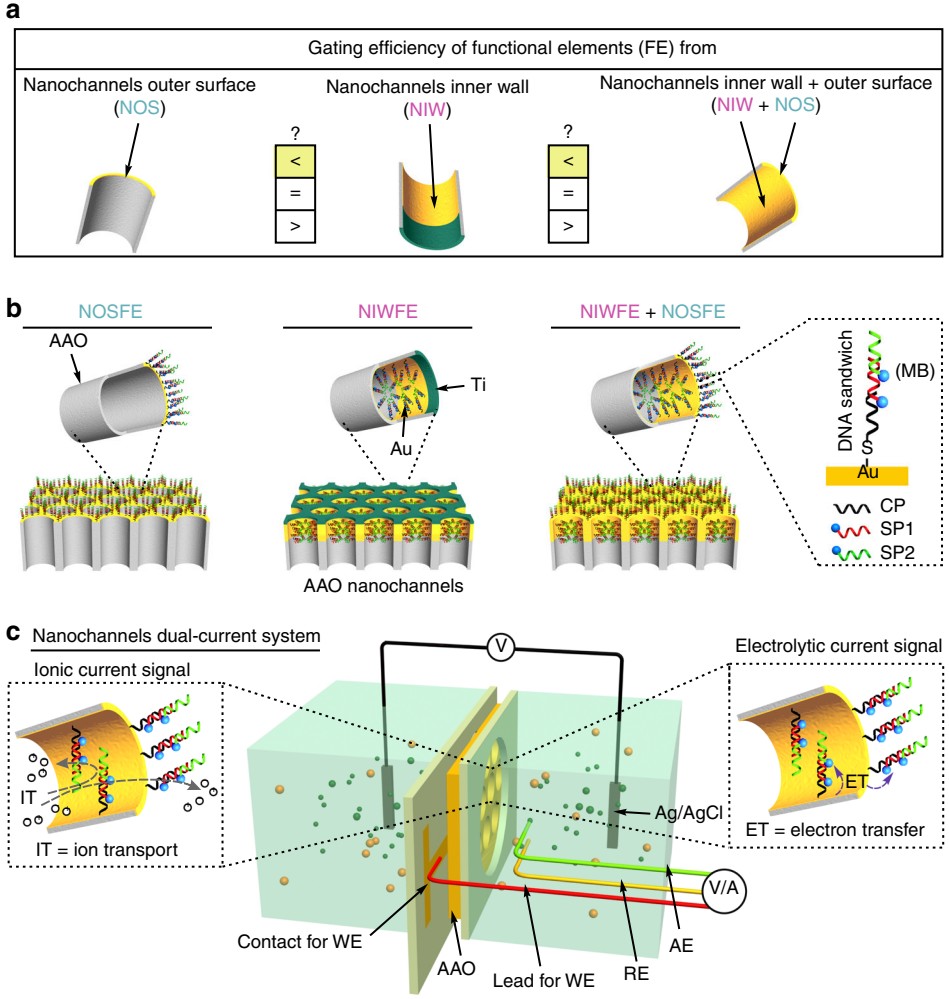

**Fig. 1** Design principle and experimental setup. **a** Graphical illustration of nanochannels functional regions (NOS, NIW, and NIW + NOS) associated with ion-gating efficiency to be investigated. **b** Cartoon presentation of DNA-based functional elements (FE) regionally assembled at different regions of metal-decorated AAO nanochannels. Shown inset is a DNA traditional sandwich (TS) tethered with double-MB electrochemical tags. CP capture probe, SP signal probe. **c** Schematic diagram of the nanochannels' dual-current setup. Ion transport (IT) and electron transfer (ET) through inner-wall and outer-surface DNA-functionalized nanochannels (NIWFE + NOSFE) are schematically presented, which simultaneously generate DNA framework mediating ionic current (IC, left panel) and electrolytic current (EC, right panel) signals through a two-electrode picoammeter and a three-electrode electrochemical analyzer, respectively. WE, working electrode; AE, auxiliary electrode; RE, reference electrode. Of note, the metal-decorated AAO nanochannels' membrane serves as WE and is in electric touch with a conducting strip on a polymethyl methacrylate (PMMA) plate that is directly connected to the electrochemical analyzer

nanochannels that cannot effectively regulate transmembrane IT process in the ATP recognition events.

**Probing of high-efficiency ion-gating systems**. We then endeavored to construct a high-efficiency ion-gating system through forming DNA superstructure (DNA SS, Supplementary Fig. 2), wherein concatenated DNA strands containing ATP aptamer and signal probes (SP1 and SP2) were repeatedly hybridized between partially complementary DNA sequences (Fig. 3a). As expected, such functionally designed molecular switches produced more noticeable EC signals (Fig. 3b–d), which benefited from the nature of DNA SS, because considerably abundant MB tags were covalently attached to the DNA framework implementing the ET process. Moreover, the DNA SS-based molecular switches displayed even higher signal-off characteristics when recognizing ATP molecules (with the EC signal change ratio of 76.2–93.0% in DNA SS vs. 62.2–74.8% in DNA TS, by comparing Fig. 3e with Fig. 2e), which was a consequence

of multiple binding events associated with ample ATP-aptamer units in this DNA architecture[41]. The comparatively lower EC change of DNA SS NIWFE relative to the two other counterparts was rationally ascribed to the limited recognition ability of ATP of the molecular switch in the confined nanospace. It should be noted that the signal-off ratio (93.0 ± 5.4%) of NIWFE + NOSFE was slightly higher than NOSFE (89.6 ± 7.8%); the exact cause, however, was unclear.

By investigation of the IC response, we found that the DNA SS-based NIWFE also possessed relatively high-efficiency ion-gating effect (with 13.6 times of IC signal change when binding to ATP, Fig. 3f). The NOSFE, however, failed to function as an effective ion gate, with only 9.4% (±1.2%) IC change after binding to ATP (Fig. 3g), similar to the above DNA TS system. These findings confirmed that the NOSFE, even in a longer and complex DNA framework, could not solely act as a "gatekeeper" to control transmembrane ion transport, probably due to the unique spatial orientation of DNA-based NOSFE framework. As shown in Fig. 3h, the IC signal change of the NIWFE + NOSFE was also

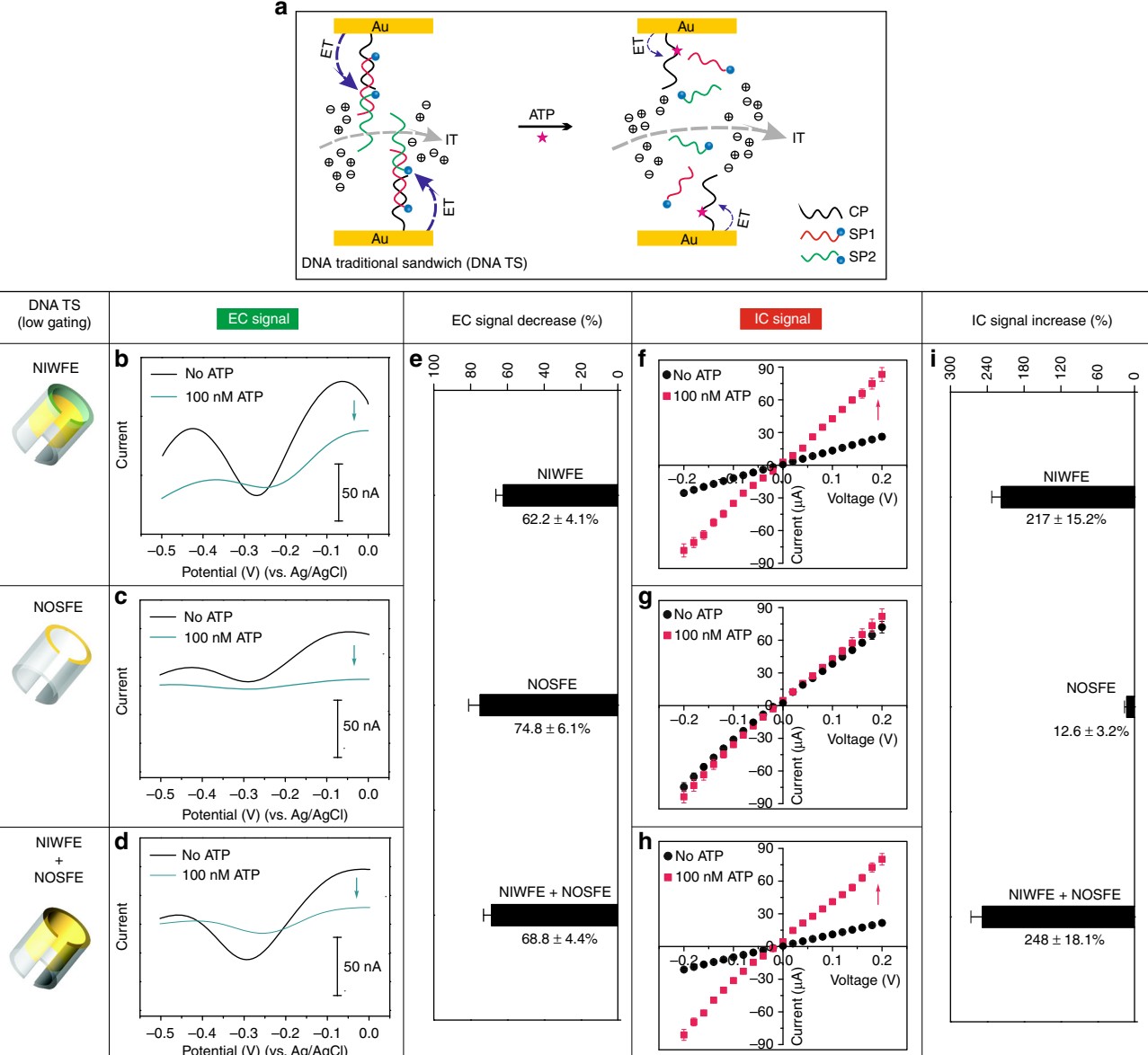

**Fig. 2** Low-efficiency ion-gating systems probed by dual-current readout. **a** Schematic of IT (dashed gray arrow) and ET (dashed blue arrow) through double labeling, ATP-responsive, DNA TS-based biomimetic nanochannels. Shown here is the disassembly process at the inner of nanochannels. **b**–**d** SWV curves of the assembled DNA TS and the disassembly by ATP at different regions of AAO nanochannels. **e** Comparison of EC signals change at the DNA TS-based ATP recognition events. **f**–**h** I–V traces of the assembled DNA TS and the disassembly by ATP at different regions of AAO nanochannels. **i** Comparison of IC signals change at the DNA TS-based ATP recognition events. Error bars represent standard deviations of independent triple experiments

increased, with 22.8 times of signal-on ratio being achieved, suggesting once again the synergistically enhanced ion-gating effect of the outer-surface probes. Importantly, such ability was rather stronger than the DNA TS system; that is, NOSFE in the DNA SS system contributed more to the overall ion-gating function (with 1.7-fold enhancement of gating efficiency) in alliance with the NIWFE (Fig. 3i), as compared to the above low-efficiency ion-gating system (with 1.1-fold enhancement of gating efficiency, Fig. 2i). We inferred that the larger dimension of DNA SS assembled at the outer surface (NOSFE) may readily interact with the inner-wall DNA molecules (NIWFE) to constitute ion gate in a collaborative way, and thus enhance the ion-gating ability. To inquire whether this invalid ion-gating behavior observed from the sole NOSFE would be affected by the pore size of nanochannels, we further examined the DNA SS-based NOSFE

using AAO nanochannels with smaller pore size (20–30 nm), only to find that the NOSFE actually cannot effectively modulate ions transport (Supplementary Fig. 5). In a series of control experiments, single labeling (SL) of TS and SS DNA architectures were employed to comprehensively interrogate the ion-gating behaviors (Supplementary Figs. 6–8). As can be seen, similar ion-gating characteristics as their counterparts of double-labeling (DL) strategy were observed, suggesting that MB electrochemical reporter in the DNA architectures would not essentially affect gating efficiency, but produce more noticeable molecular switch efficiency with regard to EC signal.

Quantitative comparison (Figs. 2i and 3i, and Supplementary Fig. 9) clearly signified that the NOSFE solely exhibited negligible ion-gating ability, but it can strengthen the ion-gating function in alliance with NIWFE, a shared phenomenon observed at the

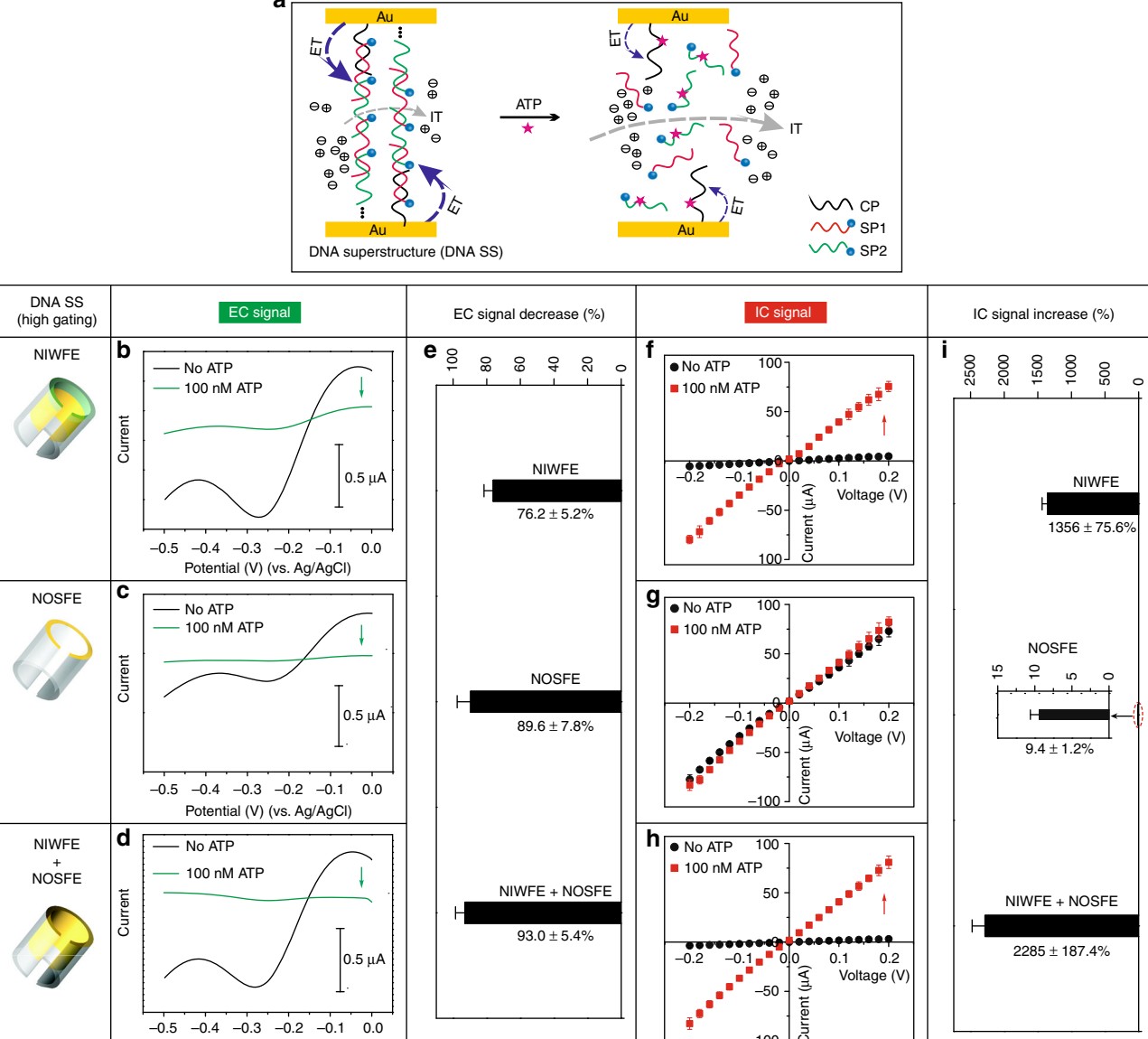

**Fig. 3** High-efficiency ion-gating systems probed by dual-current readout. **a** Schematic of IT (dashed gray arrow) and ET (dashed blue arrow) through double labeling, ATP-responsive, DNA SS-based biomimetic nanochannels. Shown here is the disassembly process at the inner of nanochannels. **b**–**d** SWV curves of the assembled DNA SS and the disassembly by ATP at different regions of AAO nanochannels. **e** Comparison of EC signals changes at the DNA SS-based ATP recognition events. **f**–**h** *I–V* traces of the assembled DNA SS and the disassembly by ATP at different regions of AAO nanochannels. **i** Comparison of IC signals changes at the DNA SS-based ATP recognition events. Error bars represent standard deviations of independent triple experiments

DNA TS and the DNA SS systems. In addition, high-efficiency gating system (DNA SS) showed more remarkably synergistic effect than low-efficiency gating system (DNA TS).

**Gating performance of regional functional elements**. We investigated the ion-gating performance of the DNA SS-based functional elements. Both EC and IC response displayed dose-dependent characteristics (Supplementary Fig. 10). The linearly altered EC signals manifested the continuously tunable ability of the established molecular switches, and the dynamic responsive range for ATP spanned 4–5 concentration magnitudes of order. Of note, the slope of the linear-fitting curves correlating to ATP concentration from 10 pM to 100 nM obtained from different regional functional elements at nanochannels was almost equal

(about $5.0 \times 10^{-4}$, Fig. 4a), demonstrating the reliability of the molecular switches. Particularly, in all the cases examined, even 10 pM concentration of ATP stimulus could give an unambiguous signal change. Biochannels commonly have strict selectivity to specific ion or molecule; our biomimetic nanochannels likewise declared this ability, since other ATP analogs with 1000 times of concentration excess could not produce substantial disturbance affecting the molecular switch and ion-gating attributes (Fig. 4b). Except for the binary signal on/off property, the functional elements of the biomimetic nanochannels can alternately behave a close–open–close attribute, corresponding to assembly–disassembly–reassembly process of the DNA-based architectures, until it lost this efficacy at the third play (Fig. 4c). It should be noted that the EC obtained at the third attempt for

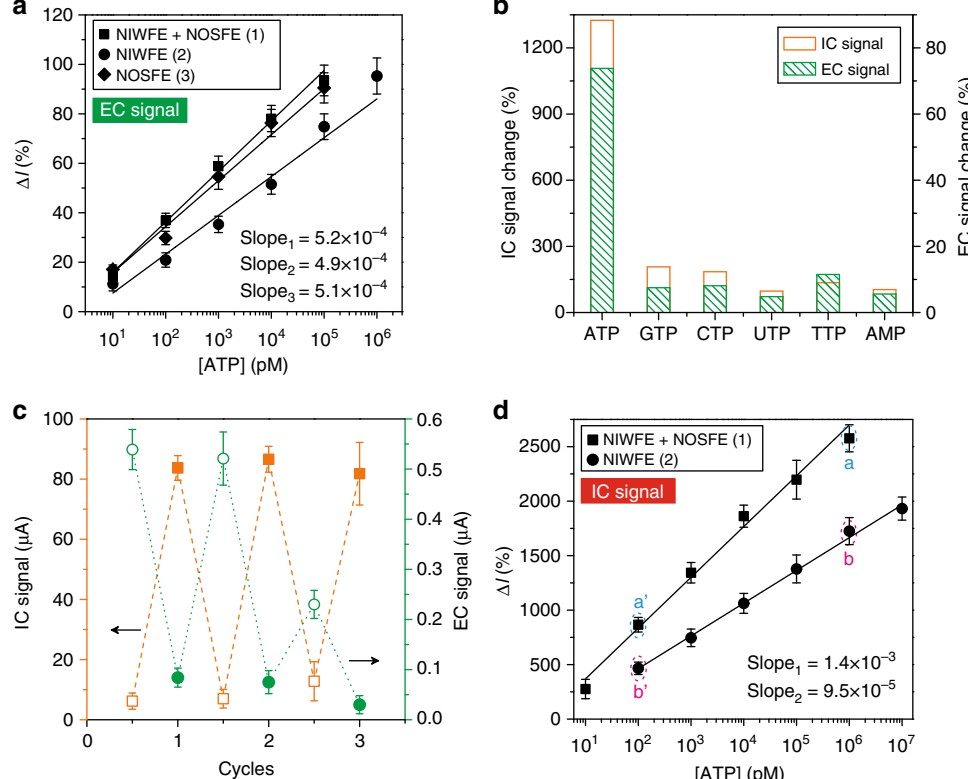

**Fig. 4** Dynamic, tunable, and reliable ion-gating behaviors. **a** Dynamic response of EC signals of the regionally constructed ATP molecular switches. **b** Specificity for ATP recognition and response. The developed biomimetic nanochannels with dual-current protocols were capable of differentiating other ATP analogs. In this case, 0.1 μM ATP and 100 μM other potential interferents were used. **c** Retrieval of molecular switch and ion-gating functions by repeatedly investigating assembly (hollow symbols) and disassembly (solid symbols) process. **d** Continuously tunable ion-gating ability of regionally assembled functional elements. The Δ*I* values notated in dashed circles were 866% (a′), 2577% (a), 469% (b′), and 1725% (b), respectively. Error bars represent standard deviations of independent triple experiments

the assembly of DNA SS was drastically attenuated, while the IC nearly remained at the original level, indicative of reserved channels blockage effect, but weakened electrochemical response, which was probably ascribed to suppressed electron-transfer ability inflicted by the repeated assembly–disassembly process that may impair the preferred orientation of DNA strands in the confined space. These findings were in line with our design of ATP-responsive, dual-signals output, intelligent biomimetic nanosystems. Not only did the DNA SS-based NIWFE effectively modulate ion transport, but also they finely guided electron transfer in a reliable and robust way. Figure 4d showed a continuously tunable ion-gating capacity within 5 magnitudes of order of ATP concentration range; higher concentration of ATP stimulus resulted in more evident IC-signal-change ratio (e.g., a > a′, b > b′). Moreover, the slope of linear-fitting curves of the NIWFE + NOSFE was greater than the NIWFE alone, and the difference between NIWFE + NOSFE with NIWFE by 1 μM ATP stimulus was larger than that observed by 0.1 nM ATP stimulus (e.g., (a–b) > (a′–b′)). All these results confirmed that NOSFE can produce synergistically enhanced ion-gating effect; the higher the gating efficiency was, the more noticeable the synergistic effect that can be obtained.

**Quantitative characterization of regional probes density**. Next to the above observations, we employed a cationic redox marker, RuHex, as electrochemical reporter, which can be electrostatically adsorbed to the phosphate moiety in the DNA molecules, and has routinely been used to calculate the assembly density of functional elements at planar gold electrode[42]. We assumed that this

electrostatic interaction might be affected by the charge and ionic strength of the nanoconfined environment[43]. In this regard, investigation of dual-current signals associated with electrostatically mediated DNA–RuHex framework (Fig. 5a) would provide insights to delve the mechanisms of regional ion-gating function of nanochannels. In this experiment, the assembly of DNA SS architecture was the same as the DNA MB system except for the absence of MB labeling (Supplementary Fig. 2), with particular efforts to probing molecule recognition and ion gating occurring at the outer-surface and inner-wall of nanochannels. Typical differential pulse voltammetry (DPV) signals acquiring from the DNA SS-based NOSFE and NIWFE were investigated, once again affirming the successful construction of our biomimetic nanochannels (Supplementary Fig. 11a). In a low-ionic strength buffer condition (10 mM Tris, pH 8.0), the assembled DNA SS–RuHex system also generated suppressed IC signal, whereas ATP-mediated disassembly resulted in 21-times IC-signal enhancement (Supplementary Fig. 11b), greater than that observed in the DNA SS–MB system (with 13-fold current change). Such improved ion-gating effect may stem from electrostatic interaction and electrokinetic effect imposing on the mobile-charged ions[15].

Electrochemical behaviors of RuHex ions entrapped in the DNA SS architecture were studied. Concentration-dependence of voltammetric response was observed. Relatively low concentration of RuHex ions produced only a characteristic adsorption peak at −0.2 to −0.3 V, whereas high concentration of RuHex ions may cause excessive coverage of the redox marker[44], producing an extra diffusion peak with peak potential around −0.15 V. DPV

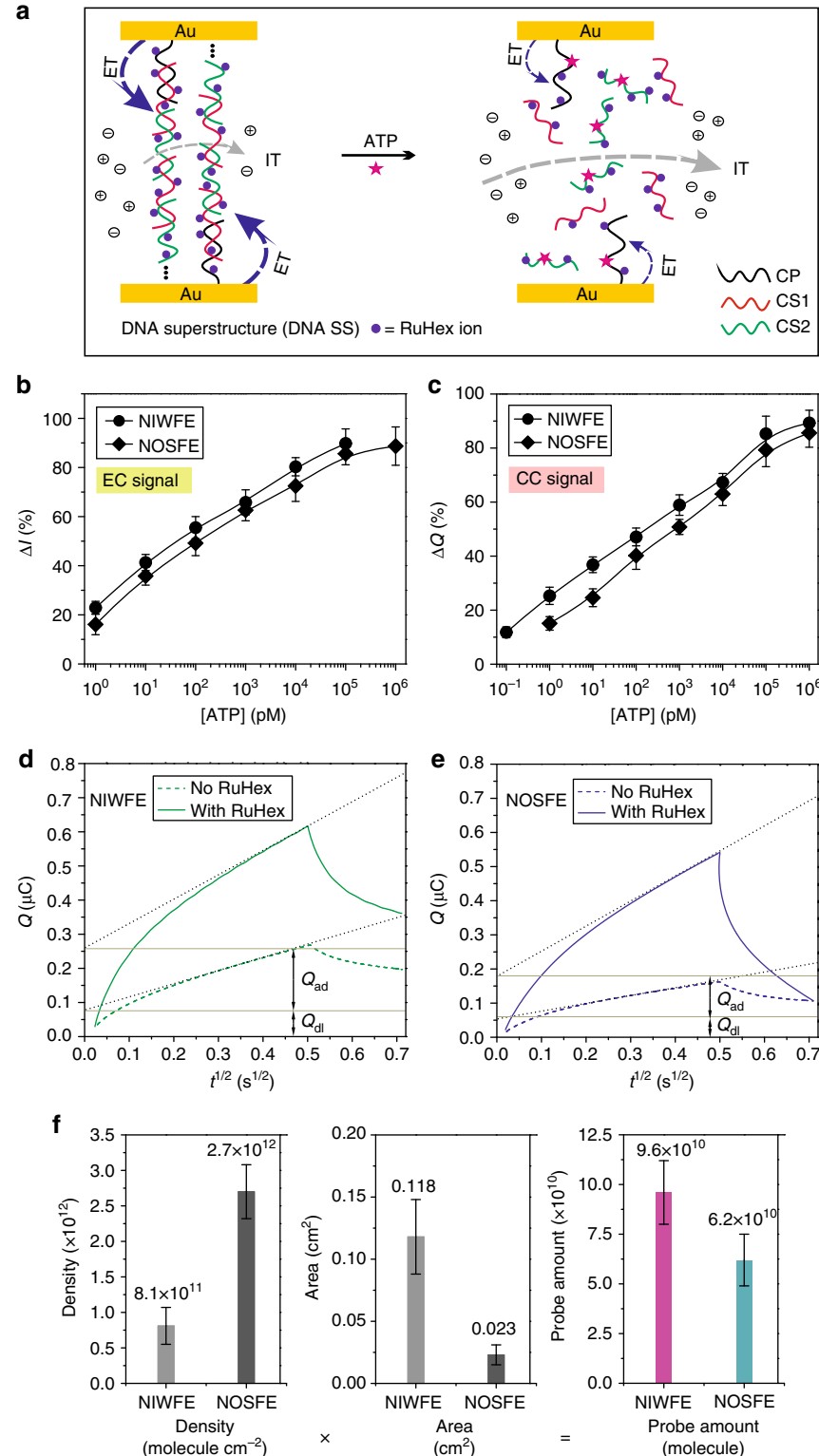

**Fig. 5** Revealing ion-gating efficiency through probes density calculation. **a** Schematic showing IT (dashed gray arrow) and ET (dashed blue arrow) through DNA SS–RuHex system. CP capture probe; CS1 complementary strand 1; and CS2 complementary strand 2 (also see Supplementary Fig. 2). **b**, **c** Dynamic response to ATP of regionally assembled DNA SS–RuHex system using differential pulse voltammetry (DPV, **b**) and chronocoulometry (CC, **c**) measurements. **d**, **e** Chronocoulometry response curves of thiolated CP + MCH monolayers with (solid line) and without (dashed line) RuHex ions adsorption at the inner wall (**d**) and outer surface (**e**) of nanochannels, respectively. Here, $Q_{ad}$ and $Q_{dl}$ represent the charge from electrochemical reduction of the adsorbed RuHex ions and the capacitive charge, respectively. **f** Comparison of DNA probe assembly density, regionally functional area, and probe molecule amount at different domains of the nanochannels. Error bars represent standard deviations of independent triple experiments

analysis further verified these features (Supplementary Fig. 12). Generally, saturated adsorption of RuHex ions in DNA backbone is a prerequisite for accurate calculations (e.g., assembly density evaluation and quantitative analysis)[42]; therefore, RuHex concentration and adsorption time were examined (Supplementary Fig. 13). Then, chronocoulometry (CC) technique was used to interrogate the molecular switch established at the DNA−RuHex architecture; the difference of charge values ($Q$) before and after ATP recognition also revealed the classic behavior of molecular switch. Moreover, linear response to ATP molecule was examined simultaneously using DPV and CC measurements (Fig. 5b, c). We found that DNA SS−RuHex-based molecular switch, relative to the DNA SS−MB system, exhibited improved signal gain, leading to more sensitive response to ATP stimulus (1−2 lower concentration magnitude of order by chronocoulometry) and much stronger switch efficiency, outperforming a traditional electrochemical switch-based ATP assay[45]. Additionally, double-layer charge and the charge due to reaction of redox molecules that diffuse to gold surface can be differentiated in the CC mode[42]. This allowed us to accurately estimate the probe density assembled at the nanochannels (Fig. 5d, e) by assuming that complete charge compensation of the DNA phosphate residues was contributed by the adsorbed redox cations (RuHex ions); this assumption was actually corroborated by chronocoulometric measurements (Supplementary Fig. 14). Accordingly, we calculated that the assembly density of the capture probe confined at the outer surface (NOSFE) and inner wall of nanochannels (NIWFE) was $2.7 \times 10^{12}$ and $8.1 \times 10^{11}$ molecules cm$^{-2}$, respectively (Fig. 5f). The relatively lower probe density of the nanochannels inner wall was primarily due to the kinetic limitation influencing on self-assembled DNA monolayer[46]; e.g., steric hindrance and electrostatic repulsion between DNA SS units[47], as compared to the bulk solution. Consistent with previous researches[48,49], more intense electrochemical signals gain can be achieved with appropriate density of DNA monolayers.

As being reminiscent to the DNA SS−MB system, this inference was yet inherently concordant. The massively adsorbed RuHex ions at the DNA SS framework were subjected to electrochemically reduced to generate EC signal through ET process; the freely existing RuHex ions (relative to the covalently bonded MB molecules) can easily escape from the inner space of nanochannels when disassembled by ATP, thus producing a sharp signal alteration and consequently exhibiting even higher switch efficiency (e.g., $89.8 \pm 5.9\%$ signal change of RuHex vs. $76.2 \pm 5.2\%$ of MB, by comparing Supplementary Fig. 11a with Fig. 3e). This also reasonably interprets the improved ion-gating efficiency of the DNA SS−RuHex system as compared to the DNA SS−MB system; that is, higher molecular switch efficiency will produce higher ion-gating efficiency as to the NIWFE. Importantly, we found that the probe amount of NOSFE and NIWFE was nearly equivalent (Fig. 5f), which was possibly a premise for the synergistically enhanced ion-gating function of NOSFE.

## Discussion

We have demonstrated a generic biomimetic nanosystem by constructing DNA-based functional elements regionally assembled at solid-state nanochannels. Using covalent labeling of MB and electrostatic adsorption of cation RuHex redox probes, dual-current signal-generation nanochannel platforms can be well established, and thus permit us to simultaneously interrogate electron transfer and ion transport process through recognition of an important biomolecule, ATP, in a confined space. Particularly, dual-current readout enables the comprehensive investigation of regional ion-gating behaviors of biomimetic nanochannels,

through which we reveal that the NOSFE cannot act as effective ion gate. By contrast, the functional elements of nanochannels' inner wall (NIWFE) play a leading role in ion-gating effect; however, the contribution differs, depending on the nature of DNA architectures (low-efficiency or high-efficiency gating systems). In addition, the NOSFE of the low-efficiency (DNA traditional sandwich) and high-efficiency (DNA superstructure) ion-gating systems can both produce synchronously enhanced ion-gating effect in alliance with NIWFE; the higher the gating efficiency is, the more noticeable the synergistic effect that can be observed. We attribute the synergistic ion-gating function to the mingled effect that may be a result of the almost equivalent distribution of NOSFE and NIWFE at nanochannels. Elaborately regulating the assembly of multifunctional polymers would gain outspread insights to biomimetic ion-gating mechanisms[20,21]. Additionally, NOSFEs simultaneously anchored at the two outer sides of a nanochannel also may be a new possibility for ion-gating research, because this can manipulate ion transport from both the entrance and the exit of the nanochannel. While electrochemically confined nanochannels exhibit the capability to mediate nucleic acid translocation, energy storage, optical enhancement, and mass transport[22–26,28,39,40,43], functional element-decorated nanochannels-on-demand will be a powerful tool for high-throughput protein characterization and single-molecule sensors. To conclude, our present integrated, smart, and dual-current readout biomimetic nanochannels are envisioned to be implicated with nanochannel logic gates, molecular electronics, biological circuits, and biosensing nanodevices.

## Methods

**Materials.** All of the chemicals were of analytical grade unless stated otherwise. Tris(hydroxymethyl) aminomethane (Tris), 6-mercapto-1-hexanol (MCH), hexaammineruthenium(III) chloride ([Ru(NH$_3$)$_6$]Cl$_3$, RuHex), and NaCl were purchased from Sigma-Aldrich (St. Louis, MO). ATP, AMP, GTP, TTP, CTP, UTP, KCl, MgCl$_2$, and tris(2-carboxyethyl) phosphine hydrochloride (TCEP) were obtained from Aladdin reagent (Shanghai, China). Single-stranded DNA sequences were synthesized and purified by Sangon Inc. (Shanghai, China). All solutions were prepared using Millipore Milli-Q water (18 MΩ cm).

**Preparation of metal-decorated AAO nanochannels.** AAO nanochannel membranes (with 40–70 nm aperture, and 60 μm thickness) were commercially obtained (Puyuan Nano, Hefei, China). Electron-beam evaporation system (Angstrom engineering, Canada) was applied for metal deposition. As for Au-decorated AAO nanochannels, a 10 or 3 nm thick Au layer was sputtered onto one side of the AAO nanochannels with deposition rate of 0.02 nm s$^{-1}$. In the case of Au/Ti decoration, a 10 nm thick Au layer was first sputtered onto one side of the AAO nanochannels (0.02 nm s$^{-1}$), followed by 5 nm thick Ti layer deposition (0.05 nm s$^{-1}$) at the same side. This process can be finely controlled by tuning the deposition rate and time, thus forming well-characteristic metal-functionalized solid-state nanochannels.

**Nanochannel characterization.** The as-prepared metal-decorated AAO nanochannel membranes were characterized by field emission scanning electron microscopy (FESEM, Nova NanoSEM 450, Netherlands) and energy dispersive X-ray spectroscopy (EDS, Oxford, UK).

**Molecule assembly.** *DNA-MB system:* Metal-decorated AAO membranes were flushed with ultrapure water and ethanol in order, and allowed to dry in a vacuum-drying system (−40 °C). To immobilize thiolated capture probe (containing ATP-aptamer sequence), the nanochannel membranes were immersed into 10 mM NaCl + 5 mM Tris (pH 7.4) buffer containing 0.1 μM CP and 100 μM TCEP for 12 h. Note that TCEP was used to cleavage disulfide bond of the thiolated capture probe, and thus facilitate the formation of self-assembled DNA monolayer through Au–S chemistry. The membranes were then rinsed with copious amounts of 10 mM Tris (pH 7.4), followed by treatment with 1 mM MCH in 10 mM Tris (pH 7.4) for 2 h to shield the residual active sites of gold interface and assist maintaining a well-organized conformation of the self-assembled DNA monolayers. For DNA superstructure (DNA SS) assembly, the nanochannel membranes were subjected to hybridization at 37 °C for 24 h in a mixture solution containing SP1 and SP2 oligonucleotides strands (0.1 μM each, in 10 mM Tris + 500 mM NaCl + 5 mM MgCl$_2$, pH 7.4). For ATP-mediated disassembly, the membranes were treated with 0.1 μM ATP at 37 °C for 3 h (in 50 mM Tris + 100 mM NaCl, pH 7.4). Finally, the membranes (both assembled and disassembled ones) were rinsed thoroughly with

10 mM Tris buffer (pH 7.4) before electrical measurements. Otherwise specific statement, 500 mM KCl solution was loaded at the two liquid chambers isolated by AAO membranes and used for current (IC and EC) measurements.

*DNA–RuHex system*: The immobilization of CP, formation of MCH monolayer, DNA hybridization (CS1/CS2, Supplementary Fig. 2), as well as ATP-mediated disassembly procedures were the same as those described above, except that a buffer containing 10 mM NaCl + 5 mM Tris + 1 mM EDTA (pH 7.4) was used to rinse the membranes when the aforementioned treatments were completed. After that, the membranes were immersed in an appropriate concentration of RuHex solution (in 10 mM Tris, pH 8.0), through which RuHex ions can electrostatically adsorb onto the negatively charged phosphate backbone in the DNA strands.

**Signal recording**. The electrical measurements were performed on customer-built polymethyl methacrylate (PMMA) electrochemical cells; wherein AAO membrane was clamped between two liquid chambers, with the metal-decorated, DNA-functionalized nanochannels being the working electrode. A polytetrafluoroethylene tube accommodated with platinum counter electrode and Ag/AgCl reference electrode was placed proximity to the membrane to form a three-electrode system. Current signal (electrolytic current, EC) acquisition was carried out on a CHI 630D electrochemical workstation (Shanghai, China). The experimental parameters were as follows, for SWV and DPV: increment of 4 mV, amplitude of 50 mV, and frequency of 10 Hz; for CC: pulse period of 250 ms and pulse width of 600 mV. Meanwhile, transmembrane IC signal was measured with a Keithley 6487 picoammeter (Keithley Instruments, Cleveland, OH) through two pieces of Ag/AgCl electrodes (0.5 mm in diameter).

**Data availability**. The data that support the findings of this study are available from the corresponding author upon reasonable request.

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

## Acknowledgements

This work was supported by the National Basic Research Program of China (973 Program, 2015CB932600), the National Key R&D Program of China (2017YFA0208000, 2016YFF0100800), the National Natural Science Foundation of China (21525523, 21722507, 21574048, 81302743, 21665004, and 21605053), the Fok Ying-Tong Education Foundation, China (151011). X.C.L. is also thankful to the financial support from China Postdoctoral Science Foundation (2015M570637), and the science fund of State Key Laboratory of Analytical Chemistry for Life Science of Nanjing University (SKLACLS1711).

## Author contributions

X.C.L. and F.X. designed the research. X.C.L. carried out the experiments. X.C.L., X.D.L., and F.X. performed data analysis. P.C.G., H.L.C., R.Z.H, and T.Y.Z. participated in nanochannels characterization. X.C.L. and F.X. wrote the paper.

## Additional information

**Competing interests:** The authors declare no competing financial interests.

