## [Peer Review File · Nature Communications]

Reviewers' comments:

Reviewer #1 (Remarks to the Author):

In the manuscript "Role of outer surface probe for regulating ion gating of nanochannels" Xianchun Li et al. report the effect of DNA grafted on the outer surface of a nanochannel on the ion gating. Both the electron transfer and ion transport are examined. The results are interesting and well presented. My questions and comments are listed below.

1. The authors claim that the contribution of functional elements on the outer surface of nanochannel is ignored in literature. This is not true. For example, the ion conductivity with and without NOSFE has been studied in Ref. 16. The NOSFE is found to enhance the conductivity in the presence of the NIWFE, in consistence with what the authors report here. In line with that work, theoretical study of sequence-designed NIWFE has been recently reported (K. Huang and I. Szeleifer, JACS, DOI: 10.1021/jacs.7b02057).
2. There are no y axis values in Fig. 2 and Fig. 3 for the EC curves. Also can the author explain the origin of the non-monotonic trend of the curves and how the EC signal decrease is defined from these curves?
3. What is the length of the inner surface? Dose NOSFE matter if the nanochannel is very long?
4. As the double strand DNA is much stiffer than the single strand counterpart, can the authors comment on the effect of such stiffness change on the ion gating?
5. The TS DNA is shorter than SS DNA. If the authors make the TS DNA longer, would it perform like the SS DNA?
6. The NOSFE are only grafted on one side of the nanochannel. What should we expect if the NOSFE are grafted on both sides?

In general, the results presented have the potential of publication but the above points need to be addressed first.

Reviewer #2 (Remarks to the Author):

This manuscript, Role of outer surface probe for regulating ion gating of nanochannels, reported on a dual-current-readout nanochannel platform to unravel regional ion gating function of biomimetic nanochannel. In conventional experiments, solid-state nanopore/nanochannel was subjected to modification with responsive molecules forming a "gate", thereby enabling to study the ion gating process. The observed ion gating is actually a whole effect from both inner wall and outer wall modification. The discrete function of regionally distributed functional molecules thus cannot be precisely identified so far, leading to a pending scientific problem in biomimetic nanopore. As the authors declared in this manuscript, the contribution of functional elements for ion gating of biomimetic nanochannel is at present unclear. To clearly address this challenge, Li and coworkers presented the novel fabrication of gating channel together with the delicate electrochemical configure in this manuscript. Upon their rational results and discussions, this manuscript gives us a clearer understanding of ion gating function of biomimetic nanochannel. More importantly, the dual-current signal mode is also an initiative effort and, can offer a complementary support to describe the

established gating systems. Unlike previous contributions, for instance, force parameter and ionic current (Keyser, U. F., et al. *Nature Physics* 2006, 2, 473), field effect transistor and ionic current (Xie, P, et al. *Nature Nanotechnology* 2012, 7, 119), fluorescence and ionic current (Huang, S., et al. *Nature Nanotechnology* 2015, 10, 986), the incorporation of ionic current and electrolytic current would add much flexibility for further nanopore-related researches.

Based on the above considerations, the authors presented a very comprehensive study both at the research strategy and data analysis on nanochannels ion gating. Due to the critical and prudent demonstrations, I hereby strongly recommend this work to be published in *Nature Communications* after address the following concerns.

1. The development for the regulation of inner wall of nanopore should be discussed, which shows advantages in single molecule sensing.
2. For what criterial the authors defined the low-efficiency and high-efficiency ion gating systems as designed in this work?
3. I think the schematic presented in Figure 2 and Figure 3 is referred to the inner-wall assembly. Give a clear description.
4. "This was probably due to the limited structure dimensions of DNA TS architectures confined at the nanochannels that cannot effectively regulate transmembrane IT process in the ATP recognition events." What is the meaning of the structure dimensions of DNA architectures?
5. How does the working electrode connect to the AAO? What's the equivalent circuit for measuring the EC in the presented experimental configure?
6. What's the thickness of gold layer or Ti layer inside the channel? Does the thickness of gold layer and/or Ti layer effect the EC and IC responses?
7. By comparison of EC and IC signals (Figure 2 and Figure 3), what is the internal connection of the dual current signals, and how to explain the inconsistent change of dual signals upon the recognition of ATP molecules?
8. What is the cause to result in more noticeable signal alteration of charge (Figure 5c) versus current of NIWFE (Figure 5b)?
9. I wonder that SL-superstructure and SL-sandwich (Figure S6 and Figure S8) was necessary to illustrate the molecular switch and ion gate efficiency.
10. Why did not the authors provide relevant data about the IC signals of NOSFE at the DNA SS-RuHex system (Figure S11b)?
11. I noticed that the adsorption equilibrium of RuHex ions was reached after one hour; this process was rather slow (Figure S13a). Please explain it.
12. The authors thought that the observed synchronous gating effect was due to the almost equally distributed FE at the outer surface and inner wall of the nanochannels. Is this a limited premise for such effect?

Reviewer #1 (Remarks to the Author):

In the manuscript "Role of outer surface probe for regulating ion gating of nanochannels" Xianchun Li et al. report the effect of DNA grafted on the outer surface of a nanochannel on the ion gating. Both the electron transfer and ion transport are examined. The results are interesting and well presented. My questions and comments are listed below.

In general, the results presented have the potential of publication but the above points need to be addressed first.

1. The authors claim that the contribution of functional elements on the outer surface of nanochannel is ignored in literature. This is not true. For example, the ion conductivity with and without NOSFE has been studied in Ref. 16. The NOSFE is found to enhance the conductivity in the presence of the NIWFE, in consistence with what the authors report here. In line with that work, theoretical study of sequence-designed NIWFE has been recently reported (K. Huang and I. Szleifer, JACS, DOI: 10.1021/jacs.7b02057).

Response: Thanks for your valuable suggestion. The role of NOSFE for ion gating in biomimetic nanochannel is actually a long-time problem to be unraveled. The research contributed by Tagliazucchi M., Rabin Y., and Szleifer I. cited as Ref. 16 in our initial manuscript (Ref. 20 in the revised manuscript) was a theoretical description of steady-state ionic fluxes through nanopores modified with polyelectrolyte chains, by assuming that they were chemically grafted at the outer surface, or outer surface + inner wall of a cylindrical nanopore. Nonequilibrium molecular theory and numerical simulation calculations were used to predict the ionic conductance behavior. This contribution was an important theoretical guideline to biomimetic nanopore design. As far as we know, no experimental efforts have been devoted to achieve the selective modification of nanopore to differentiate the ion gating capacity of functional elements (FE) regionally distributed at nanopore. This is a critical concern in the construction of biomimetic ion channel and has not yet been resolved, because the functional modification of nanopore through routine chemical grafting cannot circumvent the fact that functional molecules (e.g., organic polymers and DNA strands) will be simultaneously anchored to the surface (including outer surface and inner surface). No attempts have been made to achieve oriented

chemical grafting in the specific regions at nanopore. This has been extensively discussed in the “Introduction Section” of the manuscript. The other literature recommended by the reviewer is a very newly published paper (K. Huang and I. Szleifer. Design of multifunctional nanogate in response to multiple external stimuli using amphiphilic diblock copolymer, *J. Am. Chem. Soc.* 2017, 139, 6422); hence we have lost sight of this work during our manuscript preparation process. According to the suggestion, we add this literature in our revised manuscript (Ref. 21). For better scientific presentation, we make an appropriate amendment for this viewpoint in the revised version, presented as “*It is worth mentioning that current work is nearly focused on the contribution from the FE at nanochannels inner wall (NIWFE)^{7,13-19}; the contribution of the FE on nanochannels outer surface (NOSFE) is practically ignored. Molecular theory and simulation calculations have been proposed to predict ionic conductance and gating behavior^{20,21}; however, experimental investigations are yet unfulfilled.*” More importantly, the theoretical calculations and predictions in these two researches are in line with our present experiment results.

2. There are no y axis values in Fig. 2 and Fig. 3 for the EC curves. Also can the author explain the origin of the non-monotonic trend of the curves and how the EC signal decrease is defined from these curves?

Response: Scale bar has been given to indicate the current response amplitude, as shown in Figure 2 and Figure 3 for all the EC curves. Such illustration is a routine means in many scientific literatures to make the presentation clearer. In these combining figures, three kinds of regionally assembled FEs have been taken into account, that is, NIWFE, NOSFE, and NOSFE+NOSFE. When interrogated using EC mode, unambiguous signal change upon the binding of ATP molecules can be seen. Generally, the recognition of ATP molecules led to disassembly of the whole DNA frameworks (both at the traditional sandwich (TS), and the superstructure (SS) configurations), leaving behind surface-bound thiolated single-stranded DNA (the capture probe without MB signal reporter); as a result, the electron transfer (ET) process would be restrained and thus the current response weakened. More explicit, the EC signal decrease is defined by the current amplitude of electro-reduction peak of MB signal reporter (with peak potential of -0.25 ~ -0.3 V) through comparison of DPV curves before and after the recognition of ATP molecules.

Such EC signal change displayed non-monotonic trend, because electroactive species (here referred to MB signal reporter tethered to DNA frameworks) at DPV measurement would undergo potential-dependent and diffusion-relevant electrochemical reaction process. As for electrochemical reduction (e.g., MB molecule), the ever-decreasing electrolytic potential would prompt the electron transfer and thus enhance the electrolytic current response (more intensive current signal, referring to the absolute value), and ultimately reach the limiting diffusion current; along with the potential scan (even lower electrolytic potential), the current response starts to decline, which evolves to be a wave-shaped curve and therefore displays a non-monotonic trend at the DPV measurement, as presented in Figure 2 and Figure 3. The signal decrease is defined as $I_t/I_0 \times 100\%$ (I_t is electrolytic current measured at the disassembled state when binding to ATP molecules, and I_0 is electrolytic current measured at the assembled state; note that I_t is smaller than I_0 ; that is, EC is a signal-off mode). The data obtained from EC measurements verified that our molecular switches at nanochannels worked well.

3. What is the length of the inner surface? Dose NOSFE matter if the nanochannel is very long?

Response: The total length of the inner surface is about 60 μm (membrane thickness) for the bare AAO nanochannels. After deposition with metal layers, the inner surface length almost remained changeless (in the case of deposition of 3 nm gold, for NOSFE), suggesting that the gold layer was only located at the outer surface of the nanochannels (confirmed by the SEM and EDS testing, see Figure S1 in the Supporting Information), which enabled the assembly of DNA molecules at the outer surface via Au-S chemistry. Since the EC signal was generated from the MB-labeled DNA molecules assembled at the gold layer that constituted the regional molecular switches, we thus focused on the scale of gold layer. In the other cases, the length of inner gold surface was 0.3 μm (in the case of deposition of 10 nm-thick gold, for NIWFE + NOSFE), and 0.18 μm (in the case of deposition of 10 nm-thick gold + 5 nm-thick titanium, for NIWFE); respectively. All this information can be referred to Figure S1.

The scale of AAO nanochannels used in this work is actually very long (60 μm for the bare nanochannel), as compared to polyethylene terephthalate (12 μm , e.g., *Nano Lett.* 2009, 9, 2788; *J. Am. Chem. Soc.* 2009, 131, 8211; *J. Am. Chem. Soc.* 2010. 132, 11736;

ACS Nano 2012, 6, 3631; *J. Am. Chem. Soc.* 2014, 136, 9902) and polyimide (also 12 μm thick, *J. Am. Chem. Soc.* 2015, 137, 11976; *Adv. Mater.* 2012, 24, 6193; *J. Am. Chem. Soc.* 2009, 131, 2070; *Adv. Mater.* 2016, DOI: 10.1002/adma.201603884)-based nanochannel commercially available and widely used for biomimetic gating research. These representative contributions have been cited in our work. Besides, the length of the AAO nanochannels is at least 3~4 orders of magnitude greater than biological pores (e.g., α -hemolysin and MspA nanopore, typically having several nanometers of length). Moreover, our AAO nanochannels were much longer than the theoretical models (16 nm long; also note that $L = 256$ nm pore was modeled as long pore for theoretical calculations) as presented in Ref. 16 (*J. Am. Chem. Soc.* 2011, **133**, 17753-17763). Our findings were that NOSFE exhibited negligible ion gating ability, but it could produce synchronously enhanced effect in the presence of NIWFE when using 60 μm -long AAO nanochannels functionalized by DNA molecules. Even longer nanochannel is currently unavailable to our knowledge for biomimetic ion gating research, and it will be an interest of our future work.

4. As the double strand DNA is much stiffer than the single strand counterpart, can the authors comment on the effect of such stiffness change on the ion gating?

Response: This consideration is very meaningful. Double-stranded DNA (ds-DNA) is well recognized to be stiffer than single-stranded DNA (ss-DNA). Tarlov and coworkers indicated that the thiolated ssDNA monolayer was not a tightly packed monolayer, and that the DNA chains were not oriented perpendicular to the surface (Characterization of DNA probes immobilized on gold surfaces. *J. Am. Chem. Soc.* 1997, 119, 8916). Moreover, they found that nucleobases in ss-DNA tend to orient parallel to the gold surface in self-assembled ss-DNA films (Nucleobase orientation and ordering in films of single-stranded DNA on gold. *J. Am. Chem. Soc.* 2006, 128, 2). In this regard, ss-DNA has limited capacity to produce channel-blockage effect which is directly correlated to ion gating behavior. In contrast, the formation of ds-DNA can offset this shortage (Using self-Assembly to control the structure of DNA monolayers on gold: A neutron reflectivity study (*J. Am. Chem. Soc.* 1998, 120, 9787), since the ds-DNA (a hybridized form through complementary ss-DNA sequences) helices were observed to preferentially orient toward the substrate normal; that is, ds-DNA can orderly reorganize through base-pairing rule and

mercaptohexanol (MCH) monolayer (a similar scenario with our present work) at nanochannels, which facilitated to form more efficient ion gating system.

In this work, we designed molecular switch and ion gate by design of ds-DNA-based polymer frameworks, which contained ATP-binding aptamer sequence. This configuration was crucial to DNA-relevant gating system in nanopore/nanochannel, since it determined the open/close state and efficiency, and consequently ion flux upon the external stimulus (here referred to ATP molecule). This strategy was also widely used in the relevant research areas, for example logic gate and aptamer-based biosensing research, in non-confined environment. The significance of our ds-DNA-based molecular switch and ion gate design has 3-fold. First, the use of ds-DNA allowed us to construct the DNA framework with multiple labeling of electroactive probes (referring to the covalently-bound MB in the DNA molecules), which produced more noticeable EC signal change after recognition of ATP (by comparison of Figure 2-3 with Figure S7). If, only ss-DNA was immobilized at the specific region of the nanochannels to serve as the molecular switch, the EC recording might be excluded since it cannot provide useful information to ascertain the successful construction of the molecular switch. This is far from our dual-current signal output ion gating system. Second, ds-DNA structure (actually triple strands in the case of traditional DNA sandwich, or more in the case of the DNA superstructure) can possess larger spatial dimension capable of blocking the nanochannel to more extent. When disassembled by ATP, more remarkable change of ion flux (ionic current) can be obtained because of the change of this spatial dimension; that is, ion gate based on ds-DNA can effectively regulate the open-close behavior of our biomimetic nanochannel, which is beneficial to ion gating function close to the protein ion channels in living system. In contrast, ss-DNA may not efficiently modulate the transmembrane flux when exposure to ATP. Third, the rigidity of ds-DNA is valuable as regards mediating ET process and guiding transmembrane IT behavior, a core concept of our present work.

Taken together, ds-DNA configurations (containing double-MB labeling, referring to DNA TS; or multiple aptamer sequences, referring to DNA SS, see the manuscript) had relatively favorable rigidity than ss-DNA counterparts, which was not only beneficial to EC recording, but favorable to IC recording and thus permitted us to systematically study on regional ion gating function.

5. The TS DNA is shorter than SS DNA. If the authors make the TS DNA longer, would it perform like the SS DNA?

Response: Thanks for the referee's suggestion. We did think over this possibility. The TS-DNA framework consisted of three DNA strands containing ATP-binding aptamer sequence, and could form partly complementary structure. Both the CP and SP1 have 35-nt (containing a 27-nt ATP-binding aptamer sequence), and thus have a considerable length (*ca.* 12 nm provided that the DNA strand can stretch linearly, which has been well recognized, e.g., *J. Am. Chem. Soc.* 2012, 134, 13148). Excessive elongation of DNA strand is practically unfavorable, since it may result in structure disorder and invalid assembly of DNA framework due to the lack of rigidity as well as electrostatic repulsion and steric hindrance, which may be particularly a case in the confined space. Therefore, this may be a disadvantage to both EC and IC recordings and consequently to uncovering the biomimetic gating function. Actually, the above viewpoint was experimentally confirmed by using X-ray photoelectron, Fourier transform infrared, and near-edge X-ray absorption fine structure spectroscopy by other researchers (*J. Am. Chem. Soc.* 2006, 128, 2). Comparison of 5-nt ss-DNA and 25-nt ss-DNA assembled at gold surface indicated that orientational disorder was significantly lower in self-assembled monolayer films of 5-nt relative to 25-nt strand. The authors declared that the 25-nt strand formed mostly disordered films similar to polyelectrolyte brushes. Furthermore, simply elongating the length of DNA strands to construct DNA TS framework is also somewhat irrational, which may cause the DNA TS framework unstable due to limiting factors (e.g., electrostatic repulsion and entropy may affect the ordering of DNA strands. *J. Am. Chem. Soc.* 2006, 128, 2). This is why we call this configuration low-efficiency gating system. By contrast, DNA-SS system compensated this shortage and can form a high-efficiency gating system, and consequently enabled us to make a comprehensive investigation. Thank you for your comment. This valuable suggestion would be our future endeavor; for instance, organic polymers that can respond to pH, light, and temperature may be ideal models; or diblock copolymer grafted in the nanochannel to achieve multifunctional response feature, an elegant work recently published (*J. Am. Chem. Soc.* 2017, 139, 6422).

6. The NOSFE are only grafted on one side of the nanochannel. What should we expect if the NOSFE are grafted on both sides?

Response: Thanks a lot. If the AAO nanochannels were simultaneously modified at the both outer sides (NOSFE), the MB-labeling DNA frameworks (including TS and SS configurations) anchored at the gold surface will be isolated by the AAO base. As only one outer side of nanochannels can be electrically contacted with the electrode pad to act as working electrode responsible for monitoring the electron transfer process, we thus cannot simultaneously obtain the EC signals contributed from the both-sides modified MB-labeling DNA frameworks at the nanochannels. In other words, we cannot synchronously acquire the EC signals generated from the MB-labeling DNA frameworks that were envisioned to be simultaneously grafted at the two outer surfaces of nanochannels and serve as the “working electrodes” using our present measurement platform. This deviates from our present experimental design, a dual-current signal output biomimetic nanochannels platform. Of course, this suggestion may be a new possibility for nanochannel ion gating research; for example, chemical modification on both sides of nanochannels to explore multi-stimuli response and multifunctional gating behaviors by solely recording IC signal. Based on our present research, an expectation we can make is that the only NOSFEs from the both-sides modification similarly cannot exert effective ion gating function, and probably produce more enhanced synchronous effect when alliance with NIWFE. Such expectation is also added into the revised manuscript to manifest a more broad research prospect and will be a meaningful hint to the readers interested in nanopore territory.

Reviewer #2 (Remarks to the Author):

This manuscript, Role of outer surface probe for regulating ion gating of nanochannels, reported on a dual-current-readout nanochannel platform to unravel regional ion gating function of biomimetic nanochannel. In conventional experiments, solid-state nanopore/nanochannel was subjected to modification with responsive molecules forming a “gate”, thereby enabling to study the ion gating process. The observed ion gating is actually a whole effect from both inner wall and outer wall modification. The discrete function of regionally distributed functional molecules thus cannot be precisely identified so far, leading to a pending scientific problem in biomimetic nanopore. As the authors declared in this manuscript, the contribution of functional elements for ion gating of biomimetic

nanochannel is at present unclear. To clearly address this challenge, Li and coworkers presented the novel fabrication of gating channel together with the delicate electrochemical configure in this manuscript. Upon their rational results and discussions, this manuscript gives us a clearer understanding of ion gating function of biomimetic nanochannel. More importantly, the dual-current signal mode is also an initiative effort and, can offer a complementary support to describe the established gating systems. Unlike previous contributions, for instance, force parameter and ionic current (Keyser, U. F., et al. *Nature Physics* 2006, 2, 473), field effect transistor and ionic current (Xie, P, et al. *Nature Nanotechnology* 2012, 7, 119), fluorescence and ionic current (Huang, S., et al. *Nature Nanotechnology* 2015, 10, 986), the incorporation of ionic current and electrolytic current would add much flexibility for further nanopore-related researches.

Based on the above considerations, the authors presented a very comprehensive study both at the research strategy and data analysis on nanochannels ion gating. Due to the critical and prudent demonstrations, I hereby strongly recommend this work to be published in *Nature Communications* after address the following concerns.

1. The development for the regulation of inner wall of nanopore should be discussed, which shows advantages in single molecule sensing.

Response: Thank you for your valuable suggestion. The related literatures on nanopore single-molecule sensing have been cited in the manuscript (Ref. 26-32). As suggested, other representative work in recent years contributed by White, H. S. and Burrows, C. J. (Crown ether–electrolyte interactions permit nanopore detection of individual DNA abasic sites in single molecules. *Proc. Natl. Acad. Sci.* 2012, 109, 11504-11509. Ref. 33), Bayley, H. (Single-molecule analysis of chirality in a multicomponent reaction network. *Nat. Chem.* 2014, 6, 603-607. Ref. 34), Maglia, G. (Single-molecule analyte recognition with ClyA nanopores equipped with internal protein adaptors. *J. Am. Chem. Soc.* 2015, 137, 5793-5797. Ref. 27) have been added and discussed in the Introduction Section, which showed advantages on single molecule sensing. All these researches used nanopore-inner FE to achieve single molecule sensing with protein nanopore, such as study on individual

DNA abasic sites, single-molecule stereochemistry reaction, and single-molecule enzyme kinetics (corresponding to Ref. 33, 34, and 27; respectively).

2. For what criteria did the authors define the low-efficiency and high-efficiency ion gating systems as designed in this work?

Response: DNA sandwich-based electrochemistry is widely used for bioanalysis. Conventionally, DNA sandwich is composed of three strands acting as capture probe, recognition probe and signal probe; respectively. Similarly, in this work, we designed a DNA traditional sandwich (DNA TS) containing ATP-binding aptamer, and DNA superstructure (DNA SS) wherein concatenated DNA strands containing ATP-binding aptamer repeatedly hybridized between the partially complementary DNA strands. DNA TS framework contained a recognition unit for ATP molecule, whereas DNA SS contained multiple recognition units for ATP molecule. Therefore, we defined the DNA TS as the low-efficiency gating system and DNA SS as the high-efficiency gating system. We tested the IC and EC signals before and after binding of ATP. As expected, both the IC and EC signals at the biorecognition events related to the DNA SS framework displayed more remarkable signal change, as compared to the DNA TS framework. This affirmed the successful establishment of DNA frameworks as well as the definition of low-efficiency and high-efficiency gating systems, which allowed us to study on the regional gating function at different gating levels.

3. I think the schematic presented in Figure 2 and Figure 3 is referred to the inner-wall assembly. Give a clear description.

Response: Thank you very much. The schematics presented in Figure 2 and Figure 3 are actually referred to the inner-wall assembly in nanochannels. We have made a clear illustration in the revised manuscript.

4. "This was probably due to the limited structure dimensions of DNA TS architectures confined at the nanochannels that cannot effectively regulate transmembrane IT process in the ATP recognition events." What is the meaning of the structure dimensions of DNA architectures?

Response: As we know, ion gating effect is regulated by the open-close behavior of ion channel in living system. At biomimetic gating systems, this is commonly realized by constructing functional molecules that can respond to specific ion or molecule and effectively control the transmembrane flux. In this work, we used DNA architecture to serve as the functional molecules that respond to ATP stimulus. This design was based on the stability and manipulability of DNA molecules; typically, we can fabricate various DNA architectures via base-pairing principle. Generally, a given DNA architecture has a structure dimension, for example length and spatial scale. This is an important aspect when designing DNA-based FE in nanochannels since it determines whether the established gating system can take effect and what extent of gating efficiency it can reach. Because the DNA TS architecture assembled inner the nanochannel is shorter or smaller (length and scale of DNA stands) than the DNA SS, it can regulate the transmembrane ion flux at limited degree. Such an inference has been experimentally confirmed by the IC signal change ratios (Figure 2 and Figure 3).

5. How does the working electrode connect to the AAO? What's the equivalent circuit for measuring the EC in the presented experimental configure?

Response: The metal-deposited AAO nanochannels functioned as working electrode, which was in electrical contact with an electrode pad for EC signal measurement. Briefly, the electrode pad was fabricated on a 5 mm thick poly(methyl methacrylate) sheet (PMMA). The electrode pad was patterned using conventional lithography technique followed by magnetic sputtering deposition. First, 10-nm thick chromium (Cr) layer was presputtered on the PMMA sheet to serve as adhesive layer. Second, Au layer (*ca.* 100-nm thick) was deposited onto the Cr base. To be noted that the PMMA sheet was preserved a 3 mm diameter hole for liquid flow. Finally, the electrode pad was immobilized using epoxy glue on the prefabricated electrochemical measuring chambers, which provided electrical contact for the metal decorated AAO nanochannels. According to the suggestion, an equivalent circuit was schematically presented as follows.

Where R_s is the bulk solution resistance, R_l is the ionic conductive resistance of the interface between solid and electrolyte, R_{ct} is the charge transfer resistance, C_1 is the capacitance of outer surface, C_2 is the capacitance of inner surface, and W_o is the Warburg impedance.

6. What's the thickness of gold layer or Ti layer inside the channel? Does the thickness of gold layer and/or Ti layer effect the EC and IC responses?

Response: The metal layer was 3 nm-thick gold (for the outer-surface assembly of DNA), 10 nm-thick gold (for the outer-surface and inner-wall assembly of DNA), 10 nm-thick gold + 5 nm-thick titanium (for the inner-wall assembly of DNA); respectively. This can be achieved by controlling the deposition rate and time through electron beam evaporation technique. EDS data revealed that (1) the inner-channel gold layer thickness was negligible in the case of 3 nm-thick gold deposition, (2) the inner-channel gold layer thickness was about 0.3 μm in the case of 10 nm-thick gold deposition, and (3) the inner-channel has about 0.18 μm gold layer and 0.12 μm titanium layer in the case of 10 nm-thick gold + 5 nm-thick titanium deposition. All this information can be found in **Figure S1**.

As to the electrical signals measurements, the thickness of metal layers at the nanochannels had no obvious effect on the IC signal but affected the EC signal. The deposition thickness or mode determined the apparent area of gold layer that was directly correlated to the electrochemical reaction current (that is, electrolytic current, EC) generated from the MB-labeling DNA framework at the AAO nanochannels. Larger gold area at the nanochannels produced more noticeable EC signal; however the signal change cannot get profit when binding of ATP molecules; that is, it was not beneficial to signal change amplitude (molecular switch ratio). The preparation condition for the metal deposited AAO nanochannels presented in this manuscript was optimal and enabled to investigate the regional gating function via dual-current readout.

7. By comparison of EC and IC signals (Figure 2 and Figure 3), what is the internal connection of the dual current signals, and how to explain the inconsistent change of dual signals upon the recognition of ATP molecules?

Response: This work presented a dual-current signal output biomimetic nanochannels platform. Both the DNA TS and DNA SS frameworks were established on ATP aptamer-containing nucleotide sequences with dual MB labeling. As for the EC mode, the assembled DNA framework showed evident signal response while the disassembled form induced by ATP molecule showed remarkably attenuated signal response, which was a signal-off mode. On the contrary, as for the IC mode, the assembled DNA framework showed restrictive signal response while the disassembled form induced by ATP molecule showed remarkably enhanced signal response, which was a signal-on mode. The EC signal was modulated by the perfection of the MB-labeling DNA framework because the double-stranded DNA molecules can mediate the electron transfer (ET) process, while the IC signal was primarily affected by the transmembrane ion flux (electrostatic interaction and hydrophilic/hydrophobic feature are usually involved), in which the effective channel inner-diameter was a crucial factor. On the whole, the change trend of IC and EC signals was exactly as our expectation; the EC decreased while the IC increased. Therefore, the EC and IC signals were totally guided by the same external stimulus, ATP molecule, thus forming molecular switch and ion gate.

The somewhat inconsistent variation tendency of dual current upon the recognition of ATP molecules can be ascribed to different origins of the two kinds of signal generation, as discussed above. Additionally, we speculate that the disassembled DNA stands (both at the DNA TS and DNA SS cases) upon the recognition of ATP molecule may have a residual effect in the confined channels, which might affect the net ion flux and consequently the change amplitude of IC signal, to some extent.

8. What is the cause to result in more noticeable signal alteration of charge (Figure 5c) versus current of NIWFE (Figure 5b)?

Response: The charge signal (Q) was obtained with chronocoulometry (CC) measurement using electrochemical analyzer. As has been known, CC technique has much higher sensitivity than CV and DPV (*Anal. Chem.* 2005, 77, 6475; *Anal. Chem.* 2014, 86, 11905). In the CC mode, a large portion of RuHex ions entrapped in the heterogeneous film (here referring to the assembled DNA frameworks) are kinetically electro-inactive during “dynamic” voltammetric scans, while nearly all RuHex molecules are electroactive in the “static” chronocoulometric measurements (*J. Am. Chem. Soc.* 2006, 128, 8575). Moreover,

double-layer charge and the charge due to reaction of the adsorbed electroactive species (RuHex ions) can be differentiated in the CC mode, thus producing more sensitive and accurate electrochemical information. Therefore, it can result in more noticeable signal alteration when the DNA framework was triggered to disassemble by ATP. CC technique was commonly used for DNA-based electrochemical biosensing to improve the analytical sensitivity. More importantly, chronocoulometry measurement of RuHex ions adsorbed on DNA framework can be used to estimate the assemble density of DNA stands at the gold surface (*Anal. Chem.* 1998, 70, 4670).

9. I wonder that SL-superstructure and SL-sandwich (Figure S6 and Figure S8) was necessary to illustrate the molecular switch and ion gate efficiency.

Response: SL-superstructure and SL-sandwich were used for the control experiments. This can facilitate the illustration of much higher molecular switch efficiency of our dual MB-labeling scheme, as compared to conventional SL-DNA sandwich and superstructure. On the other, the observed ion gating effect from the SL and DL strategy was nearly not affected, further verifying the successful operation of our established DNA-based biomimetic nanochannels. This will provide more comprehensive information to the readers about our experimental design. Note that SL and DL are referring to single-labeling and double-labeling MB signal reporter, respectively.

10. Why did not the authors provide relevant data about the IC signals of NOSFE at the DNA SS-RuHex system (Figure S11b)?

Response: Thank you very much. As discussed in the manuscript, we found that NOSFE itself exhibited negligible ion gating behavior observed at the MB-labeling DNA frameworks (both the TS and SS systems). This was also the case in the DNA SS-RuHex system. Thus, the relevant data was not provided.

11. I noticed that the adsorption equilibrium of RuHex ions was reached after one hour; this process was rather slow (Figure S13a). Please explain it.

Response: The adsorption equilibrium of RuHex ions onto DNA phosphate moiety can be rapidly reached in an open space (*ca.* 5-10 minutes. *Anal. Chem.* 2014, 86, 11905). In this work, this process took place in nanoconfined space, which may impose a kinetic

limitation on the adsorption equilibrium; consequently, it took more time to reach adsorption equilibrium.

12. The authors thought that the observed synchronous gating effect was due to the almost equally distributed FE at the outer surface and inner wall of the nanochannels. Is this a limited premise for such effect?

Response: This work attempted for the first time to investigate the assembly density of DNA probes at nanochannels, with the aim to discover the origin associated with the different level of ion gating effects of regional functional elements. With CC technique that is widely used for probe density calculation at DNA-based electrochemical biosensing in unconfined condition, we here found that the probe amount of NOSFE and NIWFE was almost equally distributed in our biomimetic nanochannels. In this specific context, we therefore concluded that the experimental finding could potentially be a premise for the synchronously enhanced ion gating phenomena. Such inference was well supported by our research data and thus rational. Extensive inference without data validations cannot be offered at present.

REVIEWERS' COMMENTS:

Reviewer #1 (Remarks to the Author):

My questions and comments have been fully addressed. I support the publication of this manuscript.

Reviewer #2 (Remarks to the Author):

The revised manuscript is greatly improved and answers to all questions and remarks.

New explanation and discussions well added confirm the beautiful work on the effects of functional elements on the outer surface of confined nanochannels. This paper will intrigue a board interest in nanotechnology, sensors and biochemistry.

The reviewer recommended that the revised manuscript be published.